# Does the New Resin-Free Molten d60 Ball Have an Impact on the Velocity and Accuracy of Handball Throws?

Alfonso de la Rubia [1],*, Alexis Ugalde-Ramírez [2], Randall Gutiérrez-Vargas [2] and José Pino-Ortega [3]

1 Deporte y Entrenamiento Research Group, Departamento de Deportes, Facultad de Ciencias de la Actividad Física y del Deporte, Universidad Politécnica de Madrid, Calle Martín Fierro, 7, 28040 Madrid, Spain

2 Núcleo de Estudios para el Alto Rendimiento y la Salud (NARS), Escuela de Ciencias del Movimiento Humano y Calidad de Vida, Universidad Nacional, Heredia 40101, Costa Rica

3 Departamento de Actividad Física y Deporte, Universidad de Murcia, 30001 Murcia, Spain

* Correspondence: alfonso.delarubia@upm.es; Tel.: +34-910768013

**Abstract:** The aims of this study were (i) to examine gender differences between the Molten H3X5000 ball and the resin-free Molten d60 ball with regard to throwing velocity and accuracy according to two conditions, throwing situation and instruction received, and (ii) to analyse the player's subjective perception on throwing velocity according to ball types. The sample comprised 29 handball players (18 men and 11 women), who carried out a throwing protocol to measure velocity, accuracy, and subjective perception. The main results found significant throwing velocity differences between the new balls and traditional balls with resin in short-distance actions (7 m). Specifically, males did not perceive a loss of throwing velocity with the new ball. For target accuracy, men showed higher velocities with the traditional balls with resin and new balls than with the traditional balls with no resin. Women reached higher velocities with the new balls and the traditional balls with resin than with the traditional balls with no resin. Furthermore, throwing accuracy and effectiveness were not influenced by the ball type or throwing distance. While uneven results in relation to throwing velocity according to ball type, gender, and throwing distance were identified, the accuracy and effectiveness were not affected by the ball type. As the throws were made from further away (9 m), the impact of the new ball on the throwing velocity decreased.

**Keywords:** team sport; performance; equipment; velocity; accuracy

## 1. Introduction

Success in team sports is largely determined by the domain of certain specific skills. The most important actions in this kind of sports usually involve a high degree of ball handling and adjustment (i.e., passing), with throwing being one of them [1]. In handball, throwing ability is a crucial technical–tactical aspect because the main objective of the game is to score a goal, which can only be achieved by throwing at goal [2,3]. The fundamental requirements to perform this action efficiently and effectively are velocity—with the aim of avoiding possible defensive blocks and goalkeeper saves—and precision—trying to direct the ball to areas away from the goalkeeper [4]. The combination of these two factors has been found to be decisive in the performance [5], with a direct influence on competition [6,7].

Throwing velocity has been the most analysed criterion, considering it is a determinant of offensive success in handball [8]. The most relevant factors in this action can be classified into three groups: motion technique (i.e., [9]), physical characteristics (i.e., [10]), and motor skill (i.e., [11]). From a technical execution perspective, overarm and the inter-articular coordination of the kinetic chain involved are two of the factors with a significant impact on throwing velocity [3]. Specifically, Fradet et al. [12] showed that higher velocity in the classic overarm throw was not determined by the proximal–distal motion sequence— linear velocity gaps were observed between the joints involved—but was more dependent

on the angle and angular velocity [13]. In relation to the physical factor, the scientific literature establishes a direct or indirect relationship between throwing velocity and some anthropometric characteristics such as body breadth/body length, fat-free mass, hand size, finger length, and/or biacromial breadth [14]. Nevertheless, some research (i.e., Visnapuu and Jürimae [15]) has shown that motor coordination skills play a more relevant role than physical–anthropometric characteristics in throws at maximum velocity.

Another determining factor in the throwing action, in addition to velocity, is accuracy. The studies carried out on this subject coincide in showing that both factors had an inversely proportional relationship. For example, Indermill and Husak [16] found that the most accurate throws were made at 76% of maximum velocity, while van den Tillaar and Ettema [17] registered the most accurate throws at 85% of top speed. However, after a specific training period, it is possible to approach values of 100% maximum velocity without negatively affecting throwing accuracy [18]. This fact was corroborated by Párraga et al. [19], who identified a relationship between the level of experience and training time and the performance of powerful and accurate throws. Specifically, García et al. [20] demonstrated that in expert handball players—i.e., more trained—it was advisable to throw at the maximum velocity because accuracy, in these throwing conditions, was not significantly affected.

Both throwing factors, velocity and accuracy, can be influenced by a multitude of factors in addition to the player's level of experience. One of them is the type of instruction received by the player. In controlled tests, such as those by García et al. [20] or van den Tillaar and Ettema [4], it was shown that the type of instruction affected accuracy levels to a greater extent than velocity levels, with the latter remaining more stable in expert players than in novices. On the other hand, the throw execution technique is another relevant factor, obtaining higher velocities in the actions performed with previous running and jumping than in the static ones and with one/two feet support—i.e., standing throws [21]. However, although the velocity–accuracy relationship may balance with the experience and training levels, it behaves differently in competition. In contrast to previous studies conducted in controlled conditions, Vila et al. [22], analysing a sample of 1007 throws during official matches, found an inversely proportional relationship between effectiveness and throwing velocity, with the highest levels of accuracy being obtained at 40–50% of maximum throwing velocity.

A fundamental aspect of handball training is the practice of the throwing action with the aim of increasing the technical level. One of the main factors to consider is the type of equipment used [23]. Specifically, the ball plays a decisive role in the execution of powerful and accurate throws. Fasold et al. [24] demonstrated, in a sample of 177 young handball players, that the use of smaller and more handy balls had a positive influence on throwing velocity levels but not on accuracy levels. Thereupon, and knowing the relationship between the ball size and the hand diameter [25], a study of 104 men and women aged 5 to 33 years found that a change in ball size and weight could lead to significant variations in throwing velocity and accuracy, mostly due to changes in technical execution patterns [26]. This reality, together with the universalisation and expansion of handball, is leading the international community to change the rules of the game by proposing lighter and smaller balls with a grippier surface [27]. Thus, the International Handball Federation (IHF) has tested the performance of the new Molten d60 balls (size 2 and 3) in various competitions (e.g., World Junior Women's Championships, 2022), which have been reduced in size, and the contact surface has been replaced by a resin-free self-adhesive surface.

This study not only aims to examine the performance of the new *Molten d60* ball but also to fill the existing gap in the literature with regard to the combined evaluation of throwing velocity and accuracy in handball according to other determinants of handball performance in an integrated manner (i.e., throwing distance and instruction). Therefore, and specifically, to the best of our knowledge, the consequences of using this type of ball on the determinant actions in handball (e.g., throwing) have not yet been tested. Thus, the objectives of the present study were (i) to examine differences, in men and women separately, in throwing velocity and accuracy between the traditional ball (Molten

H3X5000) and the new ball (Molten d60) according to two contextual conditions, i.e., throwing situation (throwing technique and distance) and instruction received by the player (velocity or accuracy target), and (ii) to analyse the players' subjective perception of the different balls and conditions in relation to throwing velocity.

## 2. Materials and Methods

### 2.1. Sample

The sample for this study consisted of 29 Costa Ricans handball players (male, $n = 18$, age = $18.1 \pm 3.5$ years, height = $173.5 \pm 12.1$ cm, weight = $73.1 \pm 4.1$ kg, experience practicing handball = $4.2 \pm 3.9$ years; female, $n = 11$, age = $19.1 \pm 2.8$ years, height = $153.7 \pm 30.8$ cm, weight = $61.4 \pm 10.1$ kg, experience practicing handball = $5.1 \pm 2.3$ years). The competition level of the players was semi-professional—third competitive tier [28]. An informed consent—authorisation—was provided by all participants, or their legal representatives when they were less than 18 years old. The players reported no kind of injuries of the upper and lower extremities in the previous three years. Goalkeepers were excluded from the sample due to low use of throwing in terms of game actions [29]. The intervention protocol was conducted according to the ethics guidelines of the Declaration of Helsinki.

### 2.2. Procedures

The players carried out a non-standardised 10 min warm-up of progressively increasing intensity and acceleration consisting of dynamic running, movement, and flexibility exercises and specific passing and throwing tasks based on shoulder joint mobility. Prior to the start of the protocol, players were provided with all the information associated with the throwing test and were allowed to perform 2 pre-throws in random order to familiarise themselves with the study context [29]. The sequence of throws to the goal was randomized equally for all the participants in the same order and regardless of the throwing situation (ball, distance, and instruction) to ensure adequate rest time between throwing actions. All shots were recorded with a GoPro Hero 9 ($1920 \times 1080/16{:}9$; 120 fps) for a later review in order to establish a double observation.

#### 2.2.1. Throwing Velocity

The handball throwing velocity test was carried out using a radar gun (Stalker Sport Inc., Plano, TX, USA), with a sampling frequency of 24.2 GHz and a sensitivity of 0.045 m/s. The data were processed using the corresponding software Stalker Sport 2 (Stalker Sport Inc., Plano, TX, USA). The reliability of the system was tested by comparing the speed of rolling throws determined by the radar system with the speeds measured with photoelectric cells. The radar was positioned 1 m behind the goal net and perpendicular to the player aligned at shoulder height at the time of the shot with the aim to eliminate possible angle errors [23,30]. Throwing velocity was assessed on a $40 \times 20$ m indoor handball court in two situations, both with no defensive opposition and from a static position to avoid side effects: (1) 7 m standing throw or penalty and (2) 9 m jump throw with a run-up. In each of the two situations, the player made 2 throws with three different types of balls: (a) traditional *Molten H3X5000* ball without resin size 2 (54–56 cm circumference and 325–375 g weight, for women) or size 3 (58–60 cm circumference and 425–475 g weight, for men), (b) traditional *Molten H3X5000* ball with resin size 2 and 3, (c) new *Molten d60* ball with adhesive surface size 2 (51.5–53.5 cm circumference and 300–325 g. weight, for women), or size 3 (55.5–57.5 cm circumference and 400–425 g weight, for men) [27]. Furthermore, the players performed two sets of two throws at both distances (9 and 7 m) and with each ball (traditional ball with no resin—TBNR; traditional ball with resin—TBR; and new ball—NB) according to two different instructions (i.e., velocity objective—'throw to the centre of the goal with as much power as possible'; accuracy objective—'throw to zone 1 for right-handed players and throw to zone 3 for left-handed players') (see Figure 1). For all throws, throwing velocity was assessed. Immediately after each throwing action, the

players expressed a subjective rating of throwing velocity according to a Likert scale 1–10. Therefore, each player performed a total of 24 throws with a recovery time of 90–120 s.

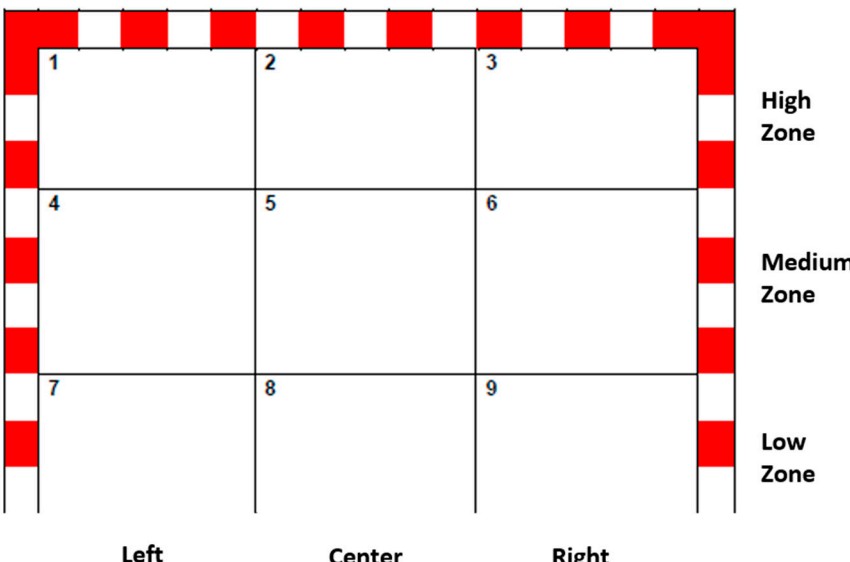

**Figure 1.** Score goal zones according to height and laterality.

### 2.2.2. Throwing Accuracy

Simultaneously, throwing accuracy was measured based on the score zone goal. Adapted from García et al. [20], the goal was divided into nine zones, resulting in nine quadrants equivalent in size and shape (0.66 m × 1 m) (Figure 1). The players were instructed to throw at the top corners (zone 1 or zone 3) according to their laterality. Every action was classified according to the following scoring system: 0 points, throws outside the goal, hitting the goalposts/crossbar, or entering the goal from any of the other eight zones (right-handed, zones 2–9; left-handed, zones 1–2 or 4–9); 1 point, shots located in zone 1—top left (right-handed)—or zone 3—top right (left-handed). Furthermore, throws that hit the posts and went into the goal were considered valid (1 point). Throwing velocity was also assessed.

### 2.2.3. Players' Subjective Perception

At the end of the throwing protocol, each player was asked about their perceived comfort level with the new ball (*Molten d60*) compared to the traditional ball (*Molten H3X5000*) using a Likert-type scale 1–10, where '1' meant no comfort and '10' meant maximum comfort.

### 2.3. Statistical Analysis

All data were collected in an Excel spreadsheet. The average throwing velocity and perceived velocity were registered with each ball, from each distance and for each target. Repeated-measures analysis of variance (ANOVA) 2 × 3 (two throwing distances x three types of balls) was performed to examine the main effect within each factor on the dependent variables (velocity and perceived velocity). ANOVAs were run separately for each gender, and their results were presented in the same way. Sphericity was tested using Mauchly's statistic. If the assumption of sphericity was not met, the Greenhouse–Geisser values were considered. Statistically significant interactions and main effects were reported for each analysis. Bonferroni post hoc tests were applied to determine differences between means for each condition. The effect size was obtained using partial eta squared ($\eta_p^2$) and was considered using the following thresholds: ≤0.12 small, ≤0.25 medium, and ≥0.26 large (Cohen, 1988). Throwing accuracy or effectiveness was analysed using a chi-square test 2 × 3 (two throwing distances × three types of balls). All statistical analyses were

performed in the Statistical Package for the Social Sciences (SPSS, IBM, v. 24.0 Chicago, IL, USA). The significance value ($\alpha$) established for all analyses was $p < 0.05$.

## 3. Results

### 3.1. Throwing Velocity

The results varied according to gender and to the instruction received. (i) 'Velocity target' (Figure 2). Significant differences were observed in men based on the type of ball ($F_{2,34} = 9.21$; $p < 0.01$; $\eta_p^2 = 0.351$), registering higher velocities in throws made with the TBR than the TBNR ($p < 0.05$) and the NB ($p < 0.01$). No differences were found according to throwing distance ($F_{1,34} = 0.17$; $p = 0.690$). The interaction effect between ball type and throwing distance was significant ($F_{2,34} = 13.45$; $p < 0.01$; $\eta_p^2 = 0.442$—Large). Multiple posterior comparisons showed that the 9 m throw velocity with the NB was higher than with the TBNR ($p < 0.05$). From 7 m, velocity was higher for throws with the TBR than with the NB ($p < 0.001$) and the TBNR ($p < 0.01$), as well as for throws with the NB compared to the TBNR ($p < 0.01$). Furthermore, higher velocities with the TBR were achieved from 7 m than from 9 m ($p < 0.05$), while higher velocities with the NB were achieved from 9 m than from 7 m ($p < 0.001$). In women, significant differences were observed according to throwing distance ($F_{1,20} = 81.94$; $p < 0.001$; $\eta_p^2 = 0.891$), with higher velocities being recorded from 9 m than from 7 m. No differences were found based on the type of ball ($F_{2,20} = 2.62$; $p = 0.098$). The interaction effect between ball type and throwing distance was significant ($F_{2,20} = 23.82$; $p < 0.001$; $\eta_p^2 = 0.704$—Large). Multiple posterior comparisons showed that the 9 m throw velocity with the NB and the TBR were higher than with the TBNR ($p < 0.01$ and $p < 0.05$, respectively). From 7 m, faster throws were recorded with the NB than with the TBR ($p < 0.05$) and the TBNR ($p < 0.001$).

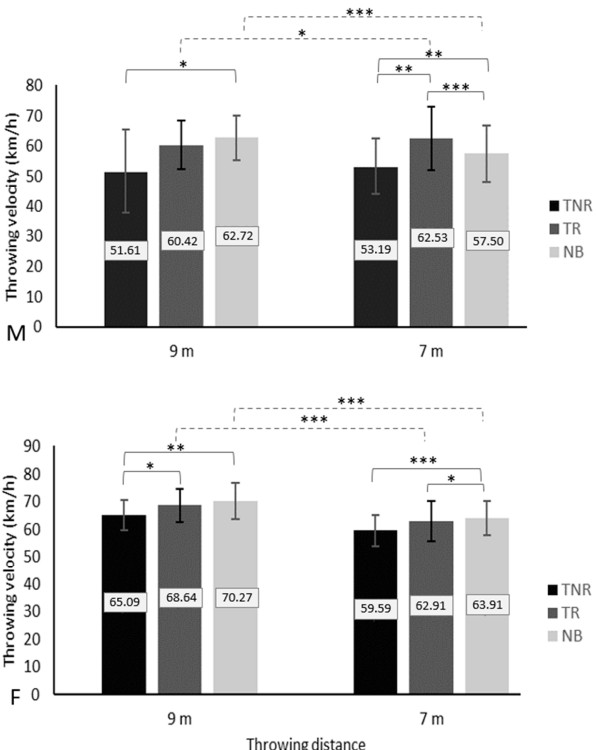

**Figure 2.** Male (M) and female (F) players' throwing velocity (radar) for velocity target according to throwing distance (7 and 9 m) and ball type (traditional ball with no resin—TBNR; traditional ball with resin—TBR; and new ball—NB). Note: continuous lines indicate significant differences between the three balls at a specific distance. Discontinuous lines indicate significant differences between the two throwing distances in a specific ball. * $p < 0.05$; ** $p < 0.01$, *** $p < 0.001$.

(ii) 'Accuracy target' (Figure 3). The interaction effect between ball type and throwing distance for both men ($F_{2,34} = 1.34$; $p = 0.277$) and women ($F_{2,20} = 0.86$; $p = 0.437$) was not significant. In men, the main effects showed significant differences according to the type of ball ($F_{2,34} = 18.07$; $p < 0.001$; $\eta_p^2 = 0.515$—Large) with higher velocities for throws made with the TBR and with the NB than with the TBNR ($p < 0.01$, both). No differences were found according to throwing distance ($F_{1,34} = 0.01$; $p = 0.987$). In women, significant differences were observed based on throwing distance ($F_{1,20} = 67.33$; $p < 0.001$; $\eta_p^2 = 0.871$—Large) with higher velocities registered from 9 m than from 7 m. Significant differences were found according to the type of ball ($F_{2,20} = 13.48$; $p < 0.001$; $\eta_p^2 = 0.574$—Large), with higher velocities being identified for throws made with the NB and the TBR than those made with the TBNR ($p < 0.01$ and $p < 0.05$, respectively).

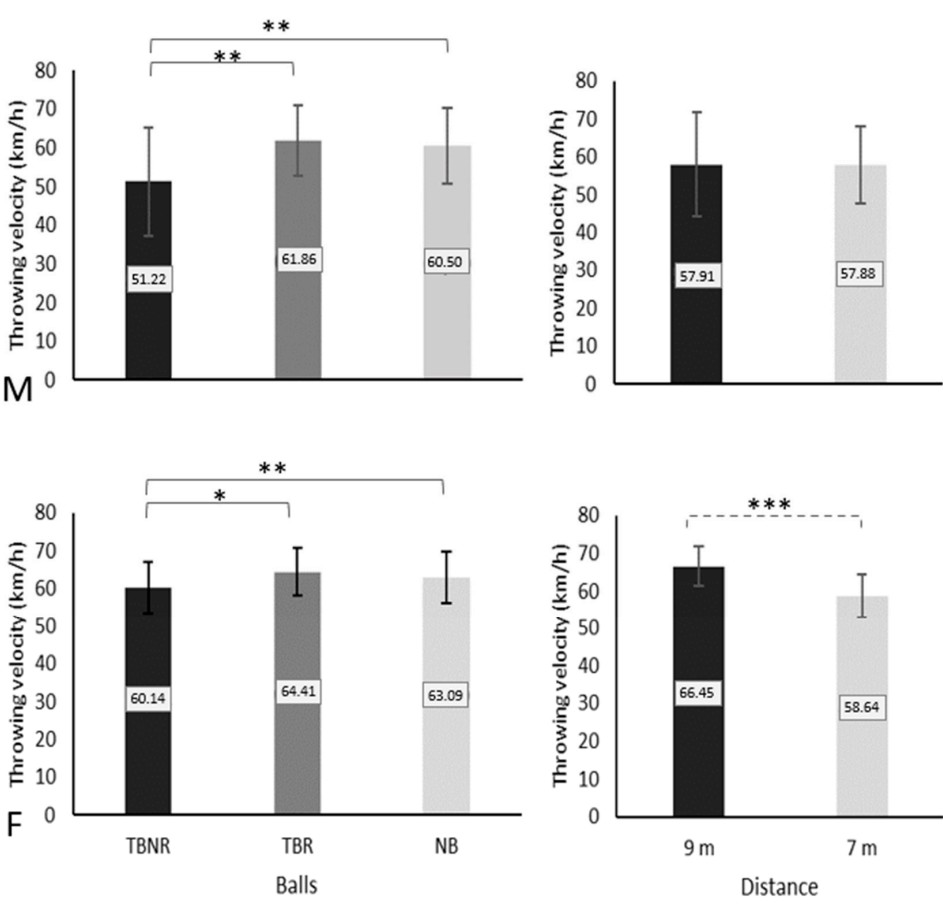

**Figure 3.** Male (M) and female (F) players' throwing velocity (radar) for accuracy target according to throwing distance (7 and 9 m) and ball type (traditional ball with no resin—TBNR; traditional ball with resin—TBR; and new ball—NB). Note: continuous lines indicate significant differences between the three balls. Discontinuous lines indicate significant differences between the two throwing distances. * $p < 0.05$; ** $p < 0.01$, *** $p < 0.001$.

With regard to the players' subjective perception associated with the throwing velocity, the results were as follows (Table 1): (i) 'Velocity target'. Significant differences were observed in men in relation to the type of ball ($F_{2,34} = 7.84$; $p < 0.01$; $\eta_p^2 = 0.316$—Large). However, in contrast to the values measured by the radar, the players reported that the NB did not give them the feeling of a loss of throwing velocity compared to the TBR ($p > 0.05$). The interaction effect between ball type and throwing distance was significant ($F_{2,34} = 3.30$; $p < 0.05$; $\eta_p^2 = 0.162$—Medium). Multiple posterior comparisons showed that throwing velocity perceptions corresponded significantly with radar measurements. The main effects, in men, showed significant differences with regard to the ball type ($F_{2,34} = 7.84$; $p < 0.01$; $\eta_p^2 = 0.316$—Large). No differences were found in women between

velocity subjective perceptions according to throwing distance ($F_{1,20}$ = 1.82; $p$ = 0.208), the ball type ($F_{2,20}$ = 0.03; $p$ = 0.968), or the interaction between both factors ($F_{2,20}$ = 1.16; $p$ = 0.333), in contrast to the radar results. (ii) 'Accuracy target'. The interaction effect between ball type and throwing distance was not significant for either men ($F_{2,34}$ = 1.15; $p$ = 0.328) or women ($F_{2,20}$ = 0.44; $p$ = 0.650). In men, the main effects showed significant differences according to the ball type ($F_{2,34}$ = 8.24; $p < 0.01$; $\eta_p^2$ = 0.326—Large), registering higher subjective velocity perceptions for throws with the NB than with the TBNR ($p < 0.01$), coinciding with the radar measurement. The women reported that distance was a determining factor in throwing velocity levels ($F_{1,20}$ = 14.11; $p < 0.01$; $\eta_p^2$ = 0.585—Large). However, female players considered the ball type to have no impact on velocity ($F_{2,20}$ = 0.83; $p$ = 0.449), in contrast to the radar measurements.

**Table 1.** Male and female players' perception level of throwing velocity (Likert scale 1–10) according to throwing distance (7 and 9 m), ball type (traditional ball with no resin—TBNR; traditional ball with resin—TBR; and new ball—NB) and instruction received (velocity and accuracy targets).

| Throws on Goal | G | Throwing Distance | TBNR | TBR | NB | F Int | $p$ | $\eta_p^2$ |
|---|---|---|---|---|---|---|---|---|
| | | | X ± SD | X ± SD | X ± SD | | | |
| Velocity Target | M | 9 m | 5.11 ± 1.21 | 6.03 ± 1.31 | 6.36 ± 1.14 | 3.30 | 0.049 | 0.162 |
| | | 7 m | 5.25 ± 1.39 | 6.50 ± 1.53 | 5.53 ± 1.39 | | | |
| | F | 9 m | 6.32 ± 1.03 | 6.18 ± 1.21 | 6.50 ± 1.02 | 1.16 | 0.333 | 0.104 |
| | | 7 m | 6.05 ± 0.96 | 6.23 ± 1.15 | 5.95 ± 1.08 | | | |
| Accuracy Target | M | 9 m | 5.17 ± 1.80 | 5.50 ± 1.26 | 6.25 ± 1.42 | 1.15 | 0.328 | 0.063 |
| | | 7 m | 5.14 ± 1.27 | 6.39 ± 1.58 | 6.42 ± 1.35 | | | |
| | F | 9 m | 6.05 ± 1.19 | 6.14 ± 0.95 | 6.05 ± 1.19 | 0.44 | 0.650 | 0.042 |
| | | 7 m | 5.18 ± 1.31 | 5.68 ± 0.87 | 5.45 ± 0.76 | | | |

Notes: G = gender; M = Male; F = Female; F Int = statistical value of the ball–distance interaction; p = significance level; $\eta_p^2$ = effect size.

### 3.2. Throwing Accuracy

In relation to throwing accuracy, the type of ball used was not associated with throwing effectiveness, neither in men from 9 m ($\chi^2$ = 0.52, $p$ = 0.772) and 7 m ($\chi^2$ = 3.92, $p$ = 0.141) nor in women from 9 m ($\chi^2$ = 0.13, $p$ = 0.935) and 7 m ($\chi^2$ = 0.86, $p$ = 0.650) (Figure 4). Furthermore, the highest effectiveness percentages were reported, in men, for the 9 m throws with the NB (39.50%) and in the 7 m throws with the TBR (40.40%). In women, higher effectiveness was reported for the 9 m throws with the NB and with the TBR (34.80%, both) and for the 7 m throws with the NB (37.90%).

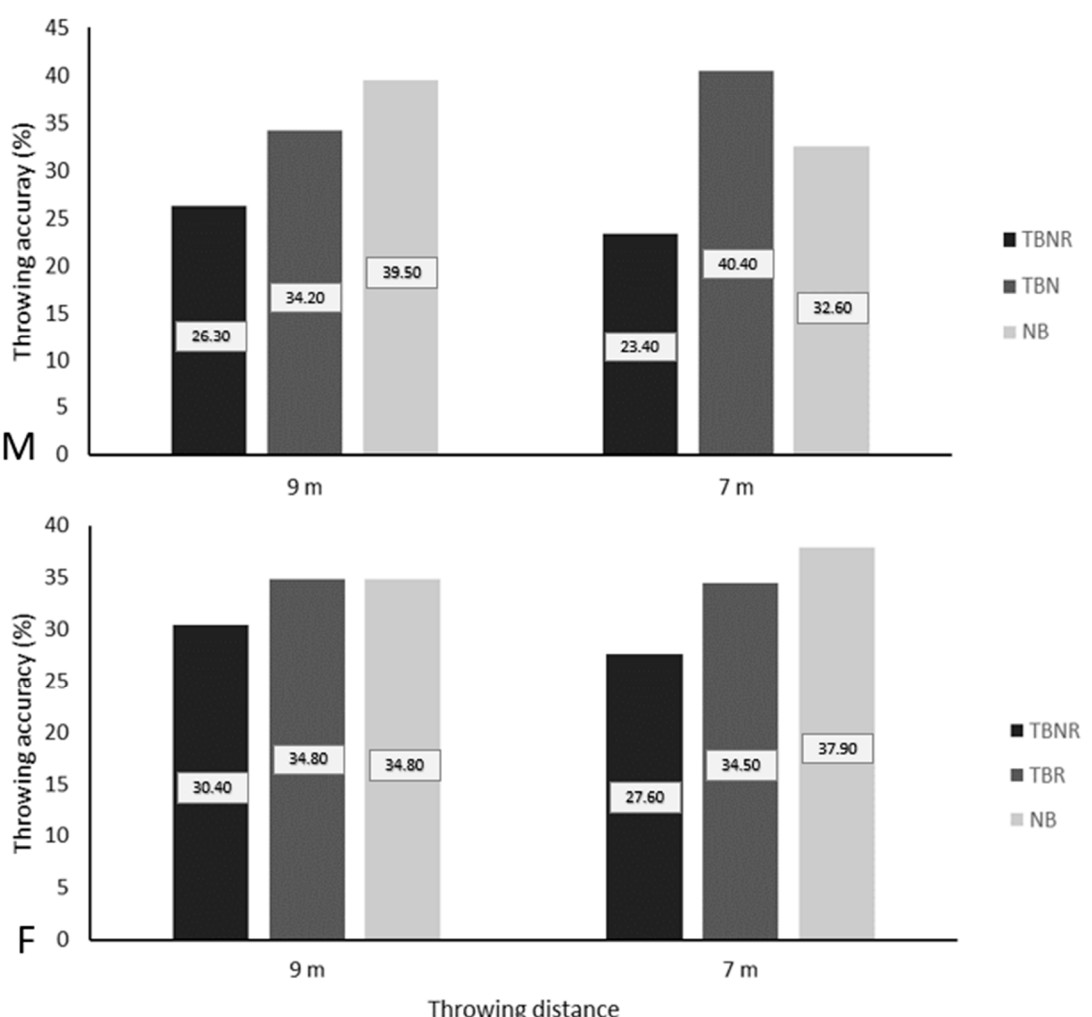

**Figure 4.** Male (M) and female (F) players' throwing accuracy for accuracy target according to throwing distance (7 and 9 m), ball type (traditional ball with no resin—TBNR; traditional ball with resin—TBR; and new ball—NB).

## 4. Discussion

This is the first scientific study that analysed the performance of the new *Molten d60* ball made with smaller dimensions and a more grippy surface that does not require the use of resin. Specifically, the study examined, firstly, the differences in the velocity and accuracy of handball throws in men and women according to three factors: ball type, throwing distance, and instruction received, and, secondly, the players' subjective perception of throwing velocity was analysed according to the aforementioned factors. The main results found significant differences between the NB *Molten d60* and the TBR with respect to throwing velocity in short-distance actions (7 m), with the effect being opposite in men versus women (i.e., female players threw faster with the NB while male players threw faster with the TBR). Specifically, males did not perceive a loss om throwing velocity in actions performed with the NB. In relation to the 'accuracy' instruction, throwing velocity did not vary between NB and TBR but was significantly lower in 'no resin' conditions—TBNR. Furthermore, throwing accuracy and effectiveness were not influenced by ball type or throwing distance.

With regard to the velocity—velocity target—more powerful throws were recorded in men with the TBR than with the NB from 7 m. Given that the use of resin has been shown to increase throwing velocity but not accuracy [31] and that the NB provides greater ball handling and adjustment capacity, it can be speculated that players opted for more accurate throws with the *Molten d60* ball to the detriment of velocity, especially in short-

distance actions (e.g., penalties). Thus, the use of this new ball would not imply high throwing velocity levels that would be achieved with the use of the resin. Furthermore, it should be added that higher throwing velocities with the NB were recorded from 9 m than from 7 m, as confirmed in previous studies (i.e., Chirosa-Ríos et al. [32], Vuleta et al. [1]). Nevertheless, these results should be considered with caution due to the high relevance of the intra-articular kinematic factor in jumping throws with a run-up, where aspects such as shoulder internal rotation, the degree of elbow flexion-extension, or pelvis rotation play a fundamental role [33].

From a short distance (7 m), the results were the opposite for women, with more powerful throws being registered with the NB *Molten d60*. This fact could suggest that the small dimensions of the NB *Molten d60* could favour high throwing velocity levels from short distances in women due to certain anthropometric characteristics such as smaller hand diameter and smaller finger size [25]. Thus, the use of a smaller and grippier ball would be beneficial for the performance in standing throws in women (without the inertia of running and/or jumping), in which there is no coadjutant acceleration of the lower body [21] and the trunk involvement is lower [34]. Therefore, NB could partially mitigate the lower throwing velocity derived from the conditional and kinematic factor in female handball players.

Examining some previous specialised literature that states that differences in throwing velocity are not exclusively the result of the kinematic factors associated with throwing technique [35]—under equal height and free fat mass conditions—it would be possible to consider other aspects such as the experience level. To this effect, recent studies have reported higher throwing velocities in professional players than in semi-professional or amateur players [1,20,32,36]. Therefore, and since in the present study female players reported high experience levels (5.1 ± 2.3 years), it would be logical to conclude that 'experience' was a determining factor in throwing velocity. Nevertheless, considering the greater training capacity of the experienced players with regard to the throwing action [3], it would be necessary to test the behaviour of the *Molten d60* ball after a training period with the aim to see if the differences according to the experience level associated with throwing velocity are maintained.

According to throwing velocity—accuracy target—no differences based in ball type and throwing distance were observed, with the exception of the 'no resin' condition, where lower velocities were recorded. Therefore, on the one hand, it can be assumed that throwing velocity tended to equalise when the player's attention was focused on throwing accuracy [17,18] and, on the other hand, that the resin played a fundamental role [31]. Similar findings were found by García et al. [20], confirming a decrease in throwing velocity under the 'accuracy' instruction, which was found to be subordinated in the player experience level, and it was more accentuated in the novice group. Moreover, several studies have shown that the use of resin has a positive influence on the more powerful throw performance, due, among other factors, to a better ball adjustment (i.e., Zapartidis et al. [25]). However, some studies (i.e., Karisic et al. [37]) found that this material is not primarily responsible for more accurate throws, although it would help in improving the technical action. This scientific evidence ratifies the results of the present study, which show that there are no significant differences in the throwing effectiveness with regard to the ball type and throwing distance, neither in men nor in women.

Players' subjective perceptions of throwing velocity showed the same trend as the radar measurements, except men did not report a slower velocity with the NB *Molten d60* than with the TBR. A larger hand size in relation to the new dimensions of the NB and no previous use of this ball could be two of the explanations for why male players did not subjectively identify throwing velocity differences between ball types. Moreover, motivational instructions or positive feedback aimed at performance improvement were able to encourage positive coping by the players with the throw carried out with the NB [38]. However, it would be necessary to replicate the subjective perception analysis with the NB in real game conditions and with the worn ball surface from use.

### 4.1. Limitations

The present study has several limitations. First, throwing performance was evaluated under non-specific conditions far from competition settings, without defensive opposition (blocking by defending players or saves by the goalkeeper) or different throwing zones. Second, we did not have an equal sample size based on gender. Third, we analysed the data according to the experience level of the male and female players due to the homogeneity existing in the sample analysed with respect to this criterion. Fourth, we did not provide the players with much practice or training time in order for the players to familiarise themselves with the use of the *Molten d60* ball.

### 4.2. Future Research Lines—Practical Applications

According to the modification of the regulations already included with regard to the official game ball—9 September 2020—and with the approval of certain national federations and leagues, it would be advisable to encourage the multidisciplinary involvement of all handball stakeholders with the aim of opening up future research lines and proposing new practical applications associated with the new *Molten d60* ball. On the one hand, an anthropometric and physical analysis of the player should be carried out in order to establish associations or predictions of physical characteristics in relation to throwing actions with the new ball. Furthermore, the use of inertial technology for the biomechanical and kinematic assessment of throw action would provide more accurate information about the behaviour of the new ball in different situations and conditions. On the other hand, it would be necessary to increase the sample sport level, especially with the aim of evaluating the throwing velocity–accuracy relationship with the new ball in highly demanding contexts under specific conditions. Finally, from a retrospective and ecological perspective, it would be useful to examine the throwing training process with the new ball in order to detect possible technical execution modifications in the action. Consequently, based on the players' subjective perception of the new ball *Molten d60*, marketing companies and manufacturers should implement ball design improvements considered relevant by the players themselves.

## 5. Conclusions

The evaluation of the new *Molten d60* ball yielded heterogeneous results with regard to throwing velocity and accuracy. While uneven results in relation to throwing velocity according to gender and throwing distance were identified (e.g., women's throws were faster from 7 m with the new ball while men's throws were faster with the traditional ball with resin), the throwing accuracy and effectiveness were not affected by the ball type. As the throws were made from farther away (i.e., from 9 m), the impact of the new ball on the throwing velocity decreased. Players' subjective perceptions helped confirm the trends assessed empirically through the radar test, except that men did not report a slower velocity with the new ball than with the traditional ball with resin.

**Author Contributions:** Conceptualization, A.d.l.R., J.P.-O., A.U.-R. and R.G.-V.; methodology, A.d.l.R. and J.P.-O.; software, A.U.-R. and J.P.-O.; validation, A.d.l.R., J.P.-O. and A.U.-R.; formal analysis, A.d.l.R., J.P.-O. and A.U.-R.; investigation, A.d.l.R., J.P.-O. and A.U.-R.; resources, A.d.l.R., J.P.-O. and A.U.-R.; data curation, A.d.l.R., J.P.-O. and A.U.-R.; writing—original draft preparation, A.d.l.R. and A.U.-R.; writing—review and editing, A.d.l.R., J.P.-O., A.U.-R. and R.G.-V.; visualization, A.d.l.R., J.P.-O. and A.U.-R.; supervision, A.d.l.R., J.P.-O. and A.U.-R.; project administration, A.d.l.R., J.P.-O., A.U.-R. and R.G.-V.; funding acquisition, J.P.-O. All authors have read and agreed to the published version of the manuscript.

**Funding:** This research received no external funding.

**Institutional Review Board Statement:** Not applicable.

**Informed Consent Statement:** Informed consent was obtained from all subjects involved in the study. Written informed consent was obtained from the patient(s) to publish this paper.

**Data Availability Statement:** The raw data supporting the conclusions of this article will be made available by the authors without undue reservation.

**Conflicts of Interest:** The authors declare no conflict of interest.

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
