# Peer review of "Does the New Resin-Free Molten d60 Ball Have an Impact on the Velocity and Accuracy of Handball Throws?"

_applsci, doi:10.3390/app13010425_

Round 1

Reviewer 1 Report

This study aims to examine whether the new resin-free Molten d60 ball has an impact on the velocity and accuracy of handball throws. The study is intriguing and relevant; however, there are some major concerns authors need to address.  

 Title:

·       I found the title too long. Maybe the second sentence is not necessary.

Abstract:

·       Try avoiding the use of abbreviations in the abstract, or at least provide the full name after the first use of the abbreviation. For example, when someone reads the abstract for the first time, he doesn’t know what NB and TBR are.

Introduction:

·       Line 45: after the word thrown, delete the extra space.

·       Other than introducing the new ball, the authors did not present an adequate rationale for this study. It should be presented more obvious where are the flaws/gaps of the previous studies examining the accuracy and velocity. And how this study might fill these gaps. I believe this study is much more than just a new ball being used (which is excellent). That`s why a more thorough rationale is needed.

Methodology:

·       Please provide the graph or image of the score zone goal. I understand that this method was previously used; however, it would be beneficial for the reader of this paper to immediately see this apparatus. This is particularly important since you explain in lines 140-141 to aim for a specific zone.

·       Regarding the previous point (because of lines 140-141), authors might consider explaining accuracy first and then velocity.

·       Authors did not explain if they randomize the trials according to the ball used, given task, and throwing distance.

·       Authors should include one paragraph regarding the data analysis. How did you analyze the data obtained from the radar gun? Via some included software or custom-made, any filters used on the raw data…?

·       Please be more specific regarding the ANOVAs. First, since you compared the two dependent variables, weren’t you supposed to use MANOVA? Or have you performed several ANOVAs? Furthermore, what does 2x3 mean? Please indicate the main effects examined (between/within factor). Also, since you mention that the results were presented separately for genders, indicate that in the statistical analysis.

Results:

·       Figure 3 – delete p<0.05 from the legend since you don’t have significant differences in this graph.

Discussion:

·       Lines 283-285: Please be cautious when discussing the specific results. You did not directly compare the results between men and women.

·       Lines 307-322. Again, this whole paragraph is regarding the differences between men and women. Answering the question of why women have higher velocities. But you did not test that. I`ve rechecked the study aims, and one of the aims is to test gender differences. However, this was not done statistically since you tested them separately.

·       Based on the previous comments, you have two options. Either perform the new statistical analysis and test the differences between men and women. Or change the study aims and discussion, so it does not rely on gender differences. Unfortunately, this paper will not be suitable for publication in all other cases.

Author Response

The reviewer can see in the following attached document the responses to the comments and suggestions made on the manuscript

Reviewer 2 Report

Small corrections are required:

Line 18 – all abbreviations should be spelled out on the 1st use. Pls do this throughout the paper.

Line 40 remove double ::

L 70 –  ‘novices oR amateurs’

Fig 1and 3. it is hard to distinguish between different types, maybe use color or different fill patterns

L277 any explanations why ‘the effect being opposite according to gender.’

L 352 ‘Third, the detailed analysis of the data according to the experience level of the male and female players.    needs explanation.

Author Response

(The authors gave the same response as above.)

Round 2

Reviewer 1 Report

Dear authors,

Thank you for your serious and scientifically accurate approach to my questions and raised issues. I`m pleased you made all corrections, including the major problem with men vs. women.

Best regards